# Urethral Bulking in the Treatment of Stress and Mixed Female Urinary Incontinence: Results from a Multicenter Cohort and Predictors of Clinical Outcomes

**DOI:** 10.3390/jcm11061569

**Published:** 2022-03-12

**Authors:** Alessandro Giammò, Paolo Geretto, Enrico Ammirati, Alberto Manassero, Luisella Squintone, Marco Falcone, Elisabetta Costantini, Giulio Del Popolo, Enrico Finazzi Agrò, Antonella Giannantoni, Vincenzo Li Marzi, Vito Mancini, Stefania Musco, Mauro Pastorello, Donatella Pistolesi, Oreste Risi, Paolo Gontero

**Affiliations:** 1Struttura Complessa di Neuro-Urologia, Città della Salute e della Scienza, Presidio CTO-USU, 10126 Torino, Italy; giammo.alessandro@gmail.com (A.G.); paolo.gere@gmail.com (P.G.); amanassero1966@gmail.com (A.M.); luisellasquintone@yahoo.it (L.S.); marco.falcone@unito.it (M.F.); paolo.gontero@unito.it (P.G.); 2Clinica Urologica, Università di Perugia, 06123 Perugia, Italy; elisabetta.costantini@unipg.it; 3Neuro-Urologia, AOU Careggi, 50134 Firenze, Italy; dpgiulio@gmail.com (G.D.P.); stefaniamusco@hotmail.com (S.M.); 4Clinica Urologica, Università di Tor Vergata, 00133 Roma, Italy; finazzi.agro@med.uniroma2.it; 5Clinica Urologica, Università di Siena, 53100 Siena, Italy; antonella.giannantoni@unisi.it; 6Clinica Urologica, AOU Careggi, 50134 Firenze, Italy; vlimarzi@hotmail.com; 7Clinica Urologica, Università di Foggia, 71122 Foggia, Italy; vito.mancini@libero.it; 8Clinica Urologica, Ospedale Sacro Cuore Don Calabria, 37024 Negrar, Italy; mauropastorello99@gmail.com; 9Clinica Urologica, Università di Pisa, 56126 Pisa, Italy; donatella.pistolesi@gmail.com; 10Clinica Urologica, Ospedale di Treviglio, 24047 Treviglio, Italy; oreste_risi@asst-bgovest.it

**Keywords:** bulkamid, female stress urinary incontinence, macroplastique, urethral bulking, urolastic

## Abstract

The aim of the present study is to analyze the outcomes of urethral bulking in the treatment of non-neurogenic female stress and mixed urinary incontinence and to assess predictors of clinical outcomes. We retrospectively included all consecutive patients affected by stress or mixed urinary incontinence and treated with urethral bulking. Outcomes were evaluated via the PGI-I questionnaire and the 24-h pad test. Between January 2010 and January 2020, we treated 216 patients (Bulkamid *n* = 206; Macro-plastique *n* = 10). The median age at surgery was 66 years (IQR 55–73.75). The median follow-up was 12 months (IQR 12–24). In total, 23.8% of patients were subjected to prior incontinence surgery, 63.8% of patients were affected by genuine stress urinary incontinence, 36.2% reported mixed urinary incontinence, whereas detrusor overactivity was confirmed in only 24.9%. The dry rate was 32.9%; nevertheless, 69.9% of patients declared themselves “very improved” or “improved” (PGI-I1-2). Low complications were observed, mostly classified as Clavien I. After univariate and multivariate analyses, the only statistically significant independent predictor of “dry” outcome was the 24 h pad test, *p* < 0.001. Urethral bulking could be proposed with more expectations of success in patients with mild urinary incontinence. Patients affected by moderate–severe incontinence are less likely to obtain clinical success; therefore, they should be carefully counselled about clinical expectations before the procedure.

## 1. Introduction

Urethral bulking (UB) is a well-established technique in the treatment of female stress and mixed urinary incontinence. Nevertheless, its relative inferiority in terms of clinical outcomes when compared to mid-urethral slings (MUS) was clearly stated by several works [1,2]. In fact, according to literature, the dry rate of UB can range from 16% to 50% despite a high satisfaction rate (up to 80% according to some studies) [2,3,4]. Furthermore, it is still not clearly stated whether functional results can be persistent over time or if the procedure needs to be repeated in order to maintain its efficacy since results from several studies are contradictory [5]. According to these observations, UB has been often regarded as a technique to be proposed mainly as a second choice to selected patients.

Nevertheless, recent reports about possible risks related to mesh surgery from several official organisms around the globe (FDA, SCHENIR) [6,7] determined rising concern about the safety of this kind of surgery in clinical activity. Therefore, the use of synthetical prostheses in the treatment of female SUI has been widely reevaluated and limited during the last years [8,9]. Hence, a renewed interest in alternative intervention for the treatment of female SUI has occurred. Among other more invasive surgical treatments (e.g., Burch’s colposuspension, autologous fascial slings, artificial sphincter implantation) urethral bulking has gained rising interest in the clinical practice as it could represent a mini-invasive and highly-safe method for the treatment of SUI.

However, given the suboptimal clinical outcomes of UB and the scarceness of solid data and of high-quality evidence, the use of UB is still largely questioned. In fact, the surgical treatment of stress urinary incontinence in females still cannot rely on solid evidence-based flowcharts, which can guide surgeons through the various techniques depending on the characteristics of patients.

The aim of the present study is to describe the outcomes and the safety profile of UB on a relatively large multicentric cohort of patients and to identify possible predictors of clinical outcomes in order to better define which patients can benefit more from a UB procedure.

## 2. Materials and Methods

The present study is an observational retrospective multicentric cohort study approved by the Institutional Ethical Committee (Intercompany Ethics Committee AOU Città della Salute e della Scienza di Torino, ID 00371/2020).

Clinical data from a multicenter cohort of consecutive patients affected by non-neurogenic stress or mixed urinary incontinence who were subjected to urethral bulking were retrospectively extrapolated from clinical records and were collected in a dedicated database. Exclusion criteria were follow-up less than 12 months, neurogenic bladder, prior major bladder surgery, male gender, pathologic findings at cystoscopy (such as bladder diverticula, undirect signs of bladder outlet obstruction, vesical stones or tumor) or incomplete data at the follow-up. Every patient underwent accurate anamnesis, physical examination and urodynamic evaluation before surgery. Physical examination included stress test, POP-Q test, perineal US scan or Q-tip test to evaluate urethral mobility. Urodynamic exams were performed according to Good Urodynamic Practice [10] as recommended by the International Continence Society (ICS).

In our clinical practice, following the EAU guidelines [11], patients with SUI are counselled about the available surgical treatment options (UB and MUS), the expected outcomes and the possible complications of each procedure. Patients who were not satisfied after urethral bulking were again counselled about the possible further therapeutic possibilities. Further options included re-UB, MUS implantation, ACT (adjustable balloons) implantation or no treatment if required by patients.

The procedure was generally performed under local anesthesia and with antibiotic prophylaxis according to institutional guidelines. A dedicated urethroscope was used, and single-use endoscopic needles were used to inject the bulking agent under the bladder neck, generally in 3 sites at the 3 o’clock position, 6 o’clock position and 9 o’clock position until a visual optimal bulking effect was achieved. Dismission from the hospital generally occurred the same day of the procedure.

Outcomes were evaluated via the Patients’ Global Impression of Improvement (PGI-I) questionnaire [12] and the 24-h pad test [13]. In the study, “dry” outcome (meaning no pads or <5 g per 24-h urine loss at pad test) was considered as a clinical success and, therefore, was included in the subsequent statistical analysis. The measure of outcomes was performed at the last follow-up, since the retrospective nature of the study and the lack of a standardized follow-up did not allow to reliantly evaluate the modifications of outcomes over time. Complications were recorded according to the Clavien–Dindo score.

The normality of variables’ distribution was tested by the Kolmogorov–Smirnov test. The categorical variables were described using frequency and percentage, and the continuous variables were described using the median and interquartile range (IQR) values. Univariate analysis was performed using the Wilcoxon test for paired data, Chi-squared test and Mann–Whiney test. Subsequently, binomial logistic regression was performed in order to assess possible independent predictors of clinical outcomes. *p* < 0.05 defined statistical significance. Statistical analysis was performed with IBM^®^ SPSS^®^ Statistics software, Version 27.0.1 (Chicago, IL, USA).

## 3. Results

In total, 306 patients subjected to urethral bulking were identified from January 2010 and January 2020 from 10 Italian centers; nevertheless, only 216 of them met the inclusion criteria and were therefore included in the study. The characteristics of the patients at the baseline are enlisted in Table 1. Bulking agents were Bulkamid (*n* = 206) and Macroplastique (*n* = 10). The median age at surgery was 66 (IQR 55–73.75). The median follow-up was 12 months (IQR 12–24). In total, 23.8% of patients were subjected to prior incontinence surgery, 63.8% of patients were affected by genuine stress urinary incontinence, 36.2% reported mixed urinary incontinence, whereas detrusor overactivity was confirmed in only 24.9%. Urethral hypermobility was found in 35.4% of patients.

The median 24 h pad test before the surgical procedure was 100 g (IQR 60–180) while, after surgery, it dropped to 35 g (IQR 10–98.75), *p* < 0.001. The dry rate at the last follow-up was 32.9%; nevertheless, 69.9% of patients declared themselves “very improved” or “improved” (PGI-I 1–2). Only 29.7% declared themselves “marginally improved” or “not improved” (PGI-I 3–4). Among complications, 11 cases of temporary urinary retention were observed that required temporary self-intermittent clean catheterization (Calvien-Dindo I), 3 cases of self-limiting hematuria (Clavien-Dindo I), 1 case of transient urethral pain, which was resolved with analgesics (Clavien-Dindo I), 1 case of urinary infection, which required prosecution of oral antibiotics (Clavien-Dindo II), and 1 case of exacerbation of urgency (although the urinary continence resulted as partially improved).

After univariate analysis, statistically significant differences were observed between the distributions of several patients’ characteristics at the baseline when stratified by the outcome “dry” (Table 2). Nevertheless, multivariate analysis remarked that the only statistically significant independent predictor of the “dry” outcome was the 24 h pad test, *p* < 0.001 (Table 3).

## 4. Discussion

In recent years, the surgical management of female SUI and MUI has undergone deep modifications, and the choice of which approach is better is still questioned. The outcomes of urethral bulking have been indagated in the past years by several studies, even if high-quality evidence is still lacking. According to the available literature, which relies mainly on two randomized-controlled clinical trials (RCTs) [14,15] and several non-randomized studies [16], the outcomes of urethral bulking are inferior when compared to the outcomes of MUS. Nevertheless, UB is often regarded as a procedure, if not the most highly effective, at least highly safe and extremely well-tolerated. The present study, consistently with the available literature, reports a dry rate of 32.9% and a satisfaction rate of 69.9%. Regarding the outcomes, it should be taken into account that, according to the most influential guidelines [11], UB has been proposed after an extensive pre-surgical counselling among techniques and most patients not willing to undergo a surgical procedure chose UB. This meant mostly elderly patients (median age 66, IQR 55–73.5) or patients that were subjected to a previous unsuccessful anti-incontinence intervention (23.8%). The outcomes could be influenced negatively by the patients’ characteristics at baseline. Moreover, the discrepancy between the objective and subjective outcomes thoroughly reflects the highly subjective perception of urinary incontinence. According to this, it is further underlined how important the preoperatory counselling and the correct patient selection is before proposing any anti-incontinence intervention.

In our case series, low complications were reported, without any high-grade complications. In fact, urethral bulking itself is not devoid of complications: some recent publications report a non-negligible incidence of high-grade complications such as the extrusion of bulking agents and urethral erosions. In particular, high-grade complications appear to be related mostly with specific bulking agents such as Macroplastique and Urolastic [17,18]. On the contrary, any report regarding serious complications following urethral bulking with Bulkamid is currently available. In our case series, most of the procedures were performed with Bulkamid as the bulking agent, which could explain the very low incidence of complications. With the aim of performing surgery as minimally invasive as possible and devoid of complication, these data could be relevant for guiding the choice of the bulking agent, even if little evidence is available.

As previously mentioned, the future goal in the treatment of female urinary incontinence should be obtaining a personalized and tailored treatment for every patient, considering the great inter-personal variability of the perception of the symptoms. Analogously, taking into account the recent concern about sling surgery, the choice of the less invasive treatment should be privileged without relinquishing the clinical effectiveness. In our study, the assessment of predictors of clinical outcomes underlined how the severity of urinary incontinence at baseline is the only independent predictor of lower success (identified as “dry” outcome), OR = 1.03 (CI 1.01–1.04, *p* < 0.001). In our analysis, dry rate in patients with mild urinary incontinence (identified as ≤50 g/24 h leakages) was 75%, while in patients with moderate or severe UI (identified as 50–200 g/24 h and >200 g/24 h leakages), dry rate was, respectively, 23% and 15%. Other patients’ characteristics such as the presence of previous anti-incontinence surgery of the presence of a low urethral closure pressure were correlated to lower objective outcomes, although they lost significance at the multivariate analysis. According to the finding of our study and similarly to some other recent works [4,19], patients with moderate-to-severe urinary incontinence should be carefully counselled about the expectations of a UB procedure and, possibly, addressed to other techniques.

We are aware that this study is not devoid of flaws. First of all, the retrospective, observational nature of the study could not allow a homogeneous follow-up; therefore, any assessment regarding the variations of the outcomes over time could be made. Moreover, the median follow-up of 12 months could not be sufficient to correctly assess long-term complications and loss of efficacy. The use of two bulking agents (Macroplastique and Bulkamid) and the characteristics of the population (high median age, high incidence of previous anti-incontinence surgery) could negatively influence the outcomes. Nevertheless, the strengths of the study are the relatively high number of patients enrolled, along with the selection of strict inclusion criteria aimed at minimizing the confounding factors. Moreover, the statistical analysis regarding the predictor of clinical outcomes could provide some practical advice and help the clinician during the counselling.

## 5. Conclusions

The present analysis confirms urethral bulking to be a highly safe procedure. Urethral bulking results in a high satisfaction rate of 69.9%. The only statistically significant predictor of clinical success is the pre operatory 24 h pad-test; therefore, it should be proposed mainly to patients with mild urinary incontinence, while patients with moderate-to-severe urinary incontinence should be carefully counselled about clinical expectations.

## Figures and Tables

**Table 1 jcm-11-01569-t001:** Baseline characteristics of the study population.

Baseline characteristics	Age at surgery Median (IQR)	66 (55–73.5)
	Previous anti-incontinence surgery *n* (%)	52 (23.8%)
	Genuine SUI *n* (%)	138 (63.8%)
	MUI *n* (%)	78 (36.2%)
	Urethral hypermobility	76 (35.4%)
	24 h pad test g/24 h Median (IQR)	100 (60–180)
Urodynamics	DO *n* (%)	54 (24.9%)
	VLPP cmH_2_O Median (IQR)	50 (35–80)
	PdetMax void cm20 Median (IQR)	12 (5–18)
	Qmax mL/s median (IQR)	20 (15–25)
	MUCP cmH_2_O Median (IQR)	50 (30–83.75)

Legend: SUI stress urinary incontinence; MUI mixed urinary incontinence; DO detrusor overactivity; VLPP Valsalva leak point pressure; PdetMax maximum detrusor pressure during voiding; Qmax maximum flow during voiding; MUCP maximum urethral closure pressure.

**Table 2 jcm-11-01569-t002:** Univariate analysis.

“Not-Dry” Outcome	
24 h pad test (g/24 h)	***p* < 0.001**
MUCP (cmH_2_O)	***p* = 0.001**
VLPP (cmH_2_O)	***p* = 0.001**
Pdet max void (cmH_2_O)	***p* = 0.001**
Previous anti-incontinence surgery (yes/no)	***p* = 0.039**
Stress test (pos/neg)	***p* = 0.020**
PVR post intervention (mL)	*p* > 0.05
Previous pelvic surgery (yes/no)	*p* > 0.05
MUI (yes/no)	*p* > 0.05
Urethral hypermobility (yes/no)	*p* > 0.05
DO (yes/no)	*p* > 0.05

**Table 3 jcm-11-01569-t003:** Multivariate analyses.

“Non Dry” Outcome		
24 h pad test (g/24 h)	***p* < 0.001**	**OR 1.03 (CI 1.01–1.04)**
MUCP (cmH_2_O)	*p* = 0.610	OR 0.99 (CI 0.97–1.01)
VLPP (cmH_2_O)	*p* = 0.990	OR 1.00 (CI 0.97–1.03)
Pdet max void (cmH_2_O)	*p* = 0.380	OR 0.97 (CI 0.90–1.04)
Previous anti-incontinence surgery (yes/no)	*p* = 0.470	OR 1.65 (CI 0.41–6.60)
Stress test (pos/neg)	*p* = 0.110	OR 2.58 (CI 0.80–8.26)

Legend: MUCP maximum urethral closure pressure; VLPP Valsalva leak point pressure; PDetMax maximum detrusor pressure during voiding; PVR post-voiding residue; MUI mixed urinary incontinence.

## Data Availability

Not applicable.

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
