# Peer review of "Urethral Bulking in the Treatment of Stress and Mixed Female Urinary Incontinence: Results from a Multicenter Cohort and Predictors of Clinical Outcomes"

_jcm, 2022, doi:10.3390/jcm11061569_

Round 1

Reviewer 1 Report

Urethral bulking in the treatment of stress and mixed female urinary incontinence: results from a multicenter cohort and predictors of clinical outcomes.

The study aimed to analyze the outcomes of urethral bulking in treating non‐neurogenic female stress and mixed urinary incontinence and assess predictors of clinical outcomes.

The success was assessed with the PGI‐I questionnaire and the 24‐hour pad test. The study group is impressive (n=216). The study is well planned, and the manuscript is well written in a very concise way. Congratulations to the Authors.

Abstract

“After univariate and multivariate…” please add the word “analysis.”

„in-continence” – please remove the hyphen

Introduction

Line 43 – please order the citations.

You state that you aimed to assess “the safety profile of UB” Why are the results not mentioned in the abstract since it was your goal?

Methods

P<=0.05 – please correct it.

Please specify the cutoffs for mild, moderate, and severe urinary incontinence with 24h pad test.

Results

You state urethral hypermobility – what were your criteria for this?

Table 1

“Not-Dry” outcome VS „ – vs??

PVR post int (ml)  - is it post intervention? It is not clear, please develop the words.

Author Response

The authors wish to thank the reviewer for his helpful and precise commentary.

We have performed the formal corrections that you suggested

We have added a statement about complications in our abstract in lines 30-31

We have better explained how we define mild, moderate or severe incontinence in lines 182-185 [“In our analysis, dry rate in patients with mild urinary incontinence (identified as <=50g/24h leakages) was 75%, while in patients with moderate or severe UI (identified as 50-200g/24h and >200g/24h leakages) dry rate was respectively 23% and 15%”] and we have also made some changes in our abstract in lines 32-36 [Urethral bulking could be proposed with more expectations of success in patients with mild-moderate urinary incontinence. Patients affected by moderate-severe incontinence are less likely to obtain clinical success, therefore they should be carefully counselled about clinical expectations before the procedure.]

Urethral hypermobility was evaluated either via the Q-tip test of via a perineal US scan (hypermobility was defined as the inclination of the urethral axis >30° from rest position)

Reviewer 2 Report

The authors describe an interesting multi-centric study about urethral bulking treatment. The results are interesting and reliable, My only concern is about relatively wide CI in multivariate analysis for previous anti-incontinence surgery and stress test. I would like authors to comment, if possible. Otherwise, statistical analysis is simple but adequate.  All conclusions are adequate.  

Author Response

We would like to thank the reviewer for his kind comment

The CI of the variables “previous anti-incontinence surgery” and “stress test” is actually relatively wide and include the value 1, therefore resulting in a non-statistically significant correlation. The difference of wideness compared to the other variables can be explained by the fact that they are categorial dichotomic variables, while the remainders are continuous variables.